# Sigma Factor Engineering in *Actinoplanes* sp. SE50/110: Expression of the Alternative Sigma Factor Gene *ACSP50_0507* (σH^As^) Enhances Acarbose Yield and Alters Cell Morphology

**DOI:** 10.3390/microorganisms12061241

**Published:** 2024-06-20

**Authors:** Laura Schlüter, Tobias Busche, Laila Bondzio, Andreas Hütten, Karsten Niehaus, Susanne Schneiker-Bekel, Alfred Pühler, Jörn Kalinowski

**Affiliations:** 1Microbial Genomics and Biotechnology, Center for Biotechnology, Bielefeld University, 33594 Bielefeld, Germany; lschluet@cebitec.uni-bielefeld.de (L.S.); schneike@cebitec.uni-bielefeld.de (S.S.-B.); 2Technology Platform Genomics, Center for Biotechnology, Bielefeld University, 33594 Bielefeld, Germany; tbusche@cebitec.uni-bielefeld.de; 3Medical School East Westphalia-Lippe, Bielefeld University, 33594 Bielefeld, Germany; 4Faculty of Physics, Bielefeld University, 33594 Bielefeld, Germany; laila.bondzio@uni-bielefeld.de (L.B.); andreas.huetten@uni-bielefeld.de (A.H.); 5Proteome and Metabolome Research, Faculty of Biology, Bielefeld University, 33594 Bielefeld, Germany; kniehaus@cebitec.uni-bielefeld.de; 6Genome Research of Industrial Microorganisms, Center for Biotechnology (CeBiTec), Bielefeld University, 33594 Bielefeld, Germany; puehler@cebitec.uni-bielefeld.de

**Keywords:** σ factor, acarbose, *Actinoplanes*, transcription, regulation, cell morphology, osmotic stress response, actinobacteria

## Abstract

Sigma factors are transcriptional regulators that are part of complex regulatory networks for major cellular processes, as well as for growth phase-dependent regulation and stress response. *Actinoplanes* sp. SE50/110 is the natural producer of acarbose, an α-glucosidase inhibitor that is used in diabetes type 2 treatment. Acarbose biosynthesis is dependent on growth, making sigma factor engineering a promising tool for metabolic engineering. ACSP50_0507 is a homolog of the developmental and osmotic-stress-regulating *Streptomyces coelicolor* σH^Sc^. Therefore, the protein encoded by *ACSP50_0507* was named σH^As^. Here, an *Actinoplanes* sp. SE50/110 expression strain for the alternative sigma factor gene *ACSP50_0507* (*sigH*^As^) achieved a two-fold increased acarbose yield with acarbose production extending into the stationary growth phase. Transcriptome sequencing revealed upregulation of acarbose biosynthesis genes during growth and at the late stationary growth phase. Genes that are transcriptionally activated by σH^As^ frequently code for secreted or membrane-associated proteins. This is also mirrored by the severely affected cell morphology, with hyperbranching, deformed and compartmentalized hyphae. The dehydrated cell morphology and upregulation of further genes point to a putative involvement in osmotic stress response, similar to its *S. coelicolor* homolog. The DNA-binding motif of σH^As^ was determined based on transcriptome sequencing data and shows high motif similarity to that of its homolog. The motif was confirmed by in vitro binding of recombinantly expressed σH^As^ to the upstream sequence of a strongly upregulated gene. Autoregulation of σH^As^ was observed, and binding to its own gene promoter region was also confirmed.

## 1. Introduction

*Actinoplanes* sp. SE50/110 is a high G + C (~71%) Gram-positive aerobic soil bacterium, featuring filamentous growth and a complex life cycle. The bacterium grows in a dense mycelium, forms motile spores [1] and has high industrial relevance as the natural producer of acarbose. Acarbose is an α-glucosidase inhibitor, used as treatment for diabetes mellitus type 2. The pseudo-tetrasaccharide acarbose consists of two parts: a maltose residue, which is α-1,4-linked to a pseudodisaccharide, consisting of aminocyclitol and deoxyribose. Its inhibitory effect is based on the complex of the unsaturated C7-aminocyclitol, composed of C7-cyclitol and aminohexose with N-glycosidic linkage [2,3]. The genome harbors 22 *acb* genes involved in acarbose biosynthesis (16) as well as its molecular modification (3) and export (3) [4,5,6,7,8,9]. The C7-aminocyclitol and aminohexose moieties are synthesized trough two different biosynthesis branches, performed by the enzymes AcbCMOLNUJR and AcbABV, respectively. Further genes are essential for extracellular modifications and transport [4,10]. Within the *acb* gene cluster, regulatory elements are scarcely known. Determination of transcription start sites (TSSs) and operon structures [11] as well as transcription dynamics and protein abundancies [12] were major steps for the broader understanding of *acb* gene regulation. In prior works, the MalR type regulator AcrC was identified to bind within the *acb* gene cluster in the intergenic region between the genes *acbE* and *acbD*, but gene deletion did not result in an increased acarbose yield [13]. To date, no transcriptional regulator capable of influence acarbose biosynthesis with an increased acarbose yield has been identified.

Since acarbose biosynthesis is growth-dependent, showing no production within the stationary growth phase [14], a focus on global transcriptional regulators with a putative influence on growth is a promising approach. For other species, it was shown that the manipulation of global transcriptional regulators can be used to achieve increased metabolite production. The manipulation can be performed using different approaches, e.g., with gene deletion, using evolved transcriptional regulators or overexpression [15,16,17]. Sigma factors were shown to be superior targets to manipulate metabolite production, as they control larger transcriptional networks. SigA overexpression successfully led to an increase in carotenoid production in *Corynebacterium glutamicum* [17], and sigma factor HrdB evolvement enabled antibiotic production increase in *Streptomyces avermitilis* [16]. Sigma factors specifically recognize promoter elements and direct core RNA polymerase (RNAP) enzyme to facilitate transcription initiation. Housekeeping sigma factors confer transcription for primary metabolism, while alternative sigma factors are active under different conditions, e.g., stress response [18,19,20,21,22], growth phase [21,23,24,25], or during secondary metabolite production [15,16,17,26,27], and these redirect RNAP to specific promoters to confer alternative gene transcription. In addition, all sigma factors compete for a limited amount of core RNAP.

Alternative sigma factors are often inactivated by binding to anti-sigma factors. Bound sigma factors can be released by direct sensing, resulting in a conformational change and release of the sigma factor [28,29], proteolysis (anti-sigma factor degeneration) [30,31] or partner-switching (inactivation by binding through anti-sigma factor antagonist) [32]. Anti-sigma factor antagonists, also named anti-anti-sigma factors, can bind anti-sigma factors, preventing the inhibition of its target sigma factor. The complex hierarchical regulatory network comprises several more steps, since anti-anti-sigma factors are activated upon diverse cellular signals [24]. Alternative sigma factors enable differential transcription, facilitating response to intra- and extracellular conditions. They play a key role in diverse stress-associated processes like heat or cold-shock response, pH shock, osmotic stress response or oxidative stress response [18,19,21,33,34,35,36]. Transcriptional-sigma-factor-mediated regulation is diverse and can differ in complexity among species. *Escherichia coli* has only seven sigma factors, one housekeeping sigma factor for essential cellular processes and six alternative sigma factors for selective differential transcription [33,34]. Other species have an even more complex regulation [37,38] with several more sigma factors involved: *Bacillus subtilis* carries 18 sigma factor genes [39,40] and *S. coelicolor* carries 65 sigma factor genes [41]. This makes the investigation challenging. Several studies elucidate σ factors networks in *S. coelicolor*, especially the stress response σB^SC^ homologs with its overlapping stress responses, connected regulatory networks and promoter cross-recognition [42,43]. The regulatory network of σB^SC^ homolog σH^SC^ is extensively characterized. σH^SC^ plays a role in osmotic stress response and morphological development [21]. Genes of the σH^SC^ regulon are associated with osmotic stress response, morphological differentiation, sporulation and aerial hyphae septation. The gene deletion strain shows impaired septation of aerial hyphae and a bald phenotype due to a decreased spore titer and reduced growth under osmotic stress [21]. σH^SC^ is regulated on a transcriptional, post-translational and protein level. Transcription of the σH^SC^ operon, coding σH^SC^ and its anti-σ factor UshX, was shown to be regulated by BldD or activated by heat shock and osmotic stress, but are also autoregulated with different promoters [21,44,45]. Further studies showed σH^SC^ regulation by proteolysis and by a partner-switching mechanism of its anti-σ factor UshX and the anti-anti-σ factor BldG. Eight anti-σ factors were identified to be able to activate BldG and fifteen σ factors were identified to interact with BldG [46,47]. This intertwined large network of regulatory partners represents the complexity of σ factor regulation, especially of the stress-associated σB homologs, and reinforces the challenge of σ factor characterization and elucidation of its regulation. This becomes even more complex with *Actinoplanes* sp. SE50/110 carrying 78 σ factor genes (Genbank: NZ_LT827010.1).

Thereby, the adaption of molecular genetic tools like integrative vectors [48], CRISPR/Cas9 [49] and homologous recombination [50] for *Actinoplanes* sp. SE50/110 as well as advances in transcriptome analysis and gene annotation enables the genetic manipulation for gene characterization and investigation of regulatory networks in this organism. We here set out to manipulate sigma factor networks in this organism to increase acarbose yield and analyze the influence of *ACSP50_0507* on the *Actinoplanes* sp. SE50/110 phenotype. Therefore, we investigated the deletion and expression mutants of a systematically selected σ factor gene with respect to growth, cell morphology and acarbose production. We further analyzed its function using transcriptomic analysis for binding motif prediction and performed binding assays. 

## 2. Materials and Methods

### 2.1. Software and Databases Used for Bioinformatic Analysis

Oligonucleotide design was performed with SnapGene 4.3 (GSL Biotech LLC, Chicago, IL, USA). EDGAR 3.0 [51], the Basic Local Alignment Search Tool [52,53,54] and CDD (Conserved Domain Database) [55,56,57,58] were used for comparative genome studies, homology prediction and functional domain search of proteins. Transcriptional analysis was performed with Readxplorer 2.2.3 and integrated DeSeq2 [59] and Improbizer [60] and WebLogo 3.7.10 [61] were used for binding motif prediction.

### 2.2. Strains, Media and Supplements

Cloning of expression and deletion plasmids was performed with *Escherichia coli* DH5αMCR, the donor strain *Escherichia coli* ET12567(pUZ8002) was used for conjugation into *Actinoplanes* sp. SE50/110, and plasmid replication for electro transformation into *Actinoplanes* sp. SE50/110 was performed using *Escherichia coli* ER2925 [62,63]. Protein expression was performed in *Escherichia coli* BL21(DE3) pLys.

*E. coli* was cultivated in liquid LB media (16 g·L^−1^ Luria/Miller broth (Carl Roth, GmbH & Co. KG, Karlsruhe, Germany) or grown on solid LB medium (supplemented with 16 g·L^−1^ agar–agar KobeI (Carl Roth, GmbH & Co. KG, Karlsruhe, Germany). The respective supplements were added to the media: apramycin (50 μg·mL^−1^), chloramphenicol (25 μg·mL^−1^), kanamycin (50 μg·mL^−1^), ampicillin (100 μg·mL^−1^).

For molecular genetic manipulation, *Actinoplanes* sp. SE50/110 was grown on solid soy flour medium (SFM: 20 g·L^−1^ soy flour, 20 g·L^−1^ mannitol, 20 g·L^−1^ agar, pH 8 adjusted with NaOH, tap water) or liquid NBS medium (11 g·L^−1^ glucose × 1H_2_O, 4 g·L^−1^ peptone, 4 g·L^−1^ yeast extract, 1 g·L^−1^ MgSO_4_·7H_2_O, 2 g·L^−1^ KH_2_PO_4_, and 4 g·L^−1^ K_2_HPO_4_) with the respective supplements.

### 2.3. Creation of Expression, Deletion and Complementation Strains

#### 2.3.1. Cloning of *ACSP50_0507* into the Integrative pSETT4tipA Expression Vector

The σ factor gene was cloned into the BbsI-linearized vector pSETT4*tipA* using Gibson Assembly [64] according to Schaffert et al. (2020) [48]. Oligonucleotides were ordered from metabion GmbH (Steinkirchen, Germany) and Sigma Aldrich (Merck KGaA, Darmstadt, Germany) (Appendix A). The target genes were amplified by Polymerase Chain Reaction (PCR) using Phusion High-Fidelity PCR Master Mix with GC Buffer (Thermo Fisher Scientific, Waltham, MA, USA). The DNA constructs were transformed into *E. coli* DH5α according to Beyer et al. (2015) and selected on solid LB media, supplemented with 50 mg L^−1^ apramycin sulfate [65]. Plates were incubated overnight at 37 °C, and the obtained clones were screened using colony PCR and agarose gel electrophoresis. Positive clones were isolated using the GeneJET Plasmid-Miniprep-Kit (Thermo Fisher Scientific, Waltham, MA, USA) and verified using Sanger sequencing by our in-house sequencing facility.

#### 2.3.2. Plasmid Transfer into *Actinoplanes* sp. SE50/110

Competent cells were obtained from 50 mL *Actinoplanes* sp. SE50/110 NBS culture, cultivated for 48 h at 140 rpm and 28 °C. After 15 min on ice, the cells were washed twice with TG buffer (10% glycerol (*v*/*v*), 1 mM Tris) and then washed in 10% (*v*/*v*) glycerin with centrifugation for 5 min at 5000× *g* and 4 °C in between. The cells were then resuspended in 3 mL of 10% (*v*/*v*) glycerin, and aliquots of 300 µL are prepared and subsequently frozen in liquid nitrogen and stored at −80 °C until further use.

Electro transformation of the pSETT4*tipA*-based expression plasmids into competent *Actinoplanes* sp. SE50/110 was performed with 100 ng expression plasmid, isolated from the methylase-deprived *E. coli* ER2925 strain, at 2500 kV, 25 µF and 200 Ω in a cuvette and incubated in 1 mL preheated CMR (10 g·L^−1^ glucose, 103 g·L^−1^ sucrose, 10.12 g·L^−1^ MgCl_2_ × 6H_2_O, 15 g·L^−1^ TSB, 5 g·L^−1^ yeast extract) at 46 °C for 6 min before incubation at 28 °C and 1100 rpm overnight. The transformation broth was then grown on solid selective solid SFM agar, supplemented with 50 mg L^−1^ apramycin sulfate, and clones were screened by colony PCR and agarose gel electrophoresis. After gDNA isolation from NBS culture using the Quick-DNA Fungal/Bacterial Miniprep (Zymo Research Corporation, Irvine, CA, USA) kit, genomic integration was verified by Sanger Sequencing and Oxford Nanopore sequencing using the MinION (Oxford, UK), and minimap2, samtools and IGV were used for mapping, indexing and visualization (Appendix A) [55,56,57].

### 2.4. Creation of the ACSP50_0507 Deletion Strain

#### 2.4.1. Construction of the Deletion Plasmid

The deletion strain was created using the CRISPR/Cas9-based vector pCRISPomyces-2 [66]. The spacer sequence was selected based on the CRISPy-web tool [67] and ordered as oligonucleotide, as well as its reversed complement, both including 4 bp overlap to the integration site of the vector (metabion GmbH, Steinkirchen, Germany and Sigma Aldrich—Merck KGaA, Darmstadt, Germany) (Appendix A). Both oligonucleotides (5 µL, 10 µM) were annealed in 30 mM of HEPES at pH 7.8 under a temperature gradient of 98 °C to 4 °C with 1 °C min^−1^ and cloned into the BsaI (NEB, Ipswich, MA, USA) linearized vector by Golden Gate Assembly following the method outlined by Cobb et al. (2015) [66]. The DNA constructs were transformed into chemo-competent *E. coli* DH5α cells according to the method outlined by Beyer et al. and selected on solid LB media supplemented with 50 mg L^−1^ apramycin sulfate [65]. Plates were incubated overnight at 37 °C and blue-white screening was used to screen positive clones, which were then isolated using the GeneJET Plasmid-Miniprep-Kit (Thermo Fisher Scientific, Waltham, MA, USA). Homologous flanks of ~1.3 kb, operating as a template for the repair of the Cas9-induced double-strand break, were amplified by PCR using Phusion High-Fidelity PCR Master Mix with GC Buffer (Thermo Fisher Scientific, Waltham, MA, USA) and cloned in the XbaI linearized vector spacer construct by Gibson Assembly [64]. Transformation was performed as mentioned prior, and colony PCR and agarose gel electrophoresis were applied for screening. Positive clones were isolated using the GeneJET Plasmid-Miniprep-Kit (Thermo Fisher Scientific, Waltham, MA, USA) and verified by Sanger sequencing by our in-house sequencing core facility.

#### 2.4.2. Conjugation

Conjugation was performed with competent *Actinoplanes* sp. SE50/110 cells and *E. coli* ET12567/pUZ8002 as a donor according to the protocol outlined by Schaffert et al. (2019) [68]. Plasmid curing of exconjugants was performed according to the method outlined by Wolf et al. (2016) [49], and the deletion was tested via PCR. Genome sequencing was performed via gDNA isolation of an NBS culture using Quick-DNA Fungal/Bacterial Miniprep (Zymo Research Corporation, Irvine, CA, USA) and ONT MinION of Oxford Nanopore (Oxford, UK) kits, and minimap2, samtools and IGV were used for mapping, indexing and visualization (Appendix A) [69,70,71].

### 2.5. Cultivation of Actinoplanes sp. SE50/110 Strains

For the pre-culture, 30 mL of NBS was inoculated from 200 µL of glycerol stock. After cultivation for 72 h at 28 °C and 140 rpm, 0.5 mL of culture was passaged to 30 mL of new media and cultivated for another 48 h. Shake flask cultivation was performed using 250 mL Erlenmeyer baffled cell culture flasks (Corning Inc., Corning, NY, USA) for 50 mL of culture at 28 °C and 140 rpm in maltose minimal medium ((1) 72.06 g·L^−1^ maltose·1H_2_O, (2) 5 g·L^−1^ (NH_4_)_2_SO_4_, 0.184 g·L^−1^ FeCl_3_·4H_2_O, 5.7 g·L^−1^ Na_3_C_6_H_5_O_7_·2H_2_O, 1 g·L^−1^, 200 µL trace elements (stock concentration: 1 μM CuCl_2_, 50 μM ZnCl_2_, 7.5 μM MnCl_2_), (3) MgCl_2_·6H_2_O, 2 g·L^−1^ CaCl_2_·2H_2_O, and (4) 5 g·L^−1^ each K_2_HPO_4_ and KH_2_PO_4_). Therefore, four medium components were separately prepared and autoclaved, whereas component (1) was sterile filtrated. Inoculation was performed using 30 mL of pre-culture, which was harvested by centrifugation at 5000 rpm for 5 min and washed twice with maltose minimal medium. The cell wet weight was measured, and the cells were diluted to 50 g mL^−1^, of which 0.5 mL was used to inoculate 50 mL of maltose minimal medium. Cultivation was performed at 140 rpm and 28 °C. Growth was determined by a cell dry weight of 2 × 1 mL of culture sample in pre-weight 1.5 mL reaction tubes, after harvest by centrifugation at 14,000 rpm for 5 min and washing with H_2_O (MilliQ) twice and a 48 h drying process at 60 °C.

### 2.6. Scanning Electron Microscopy (SEM)

*Actinoplanes* sp. SE50/110 strains were cultivated in liquid maltose minimal medium for 8 days. 1 mL cell culture sample was washed twice with 0.5 mL PBS with centrifugation at 1500 rpm for 1 min in between to remove remaining media. For purification, 0.5 mL cells solution was then transferred in a 5 mL glass vial and mixed with 5 mL of fixation reagent (2.5% (*v*/*v*) glutaraldehyde (Serva) in 100 mM phosphate buffer (pH 7.2) and incubated for 1 h on ice. Afterwards, the fixation solution was removed, and the samples were incubated in 5 mL of 100 mM phosphate buffer (pH 7.2) on ice for an additional 20 min. The solution was removed, and cells were then treated with 2-propanol (technical) in ascending concentrations, each for 20 min. The concentrations that were used were 30%, 40%, 50%, 60%, 70%, 80%, 90%, 98% of 2-propanol (technical) in 100 mM phosphate buffer (*v*/*v*). Lastly, the cells were incubated twice in 100% EtOH (HPLC grade) for 20 min. A measure of 100 µL of sample was then added to a 0.2 µm polycarbonate membrane filter (Poretics, OSMONICS, Livermore, CA, USA), dried and taped on a 12.7 mm × 3.1 mm carrier pin (MT7523, SMCAM, Science Services GmbH, Munich, Germany), before coating with gold for 150 s, at 30 mA (SCD 005 Sputter coater, BAL-TEC) under vacuum. A Hitachi S-450 Scanning Electron Microscope (High-Technologies Ltd., Tokyo, Japan) was used at 15 kV.

For the cell surface investigation, samples were alternative coated with a ~10 nm layer of ruthenium and imaged using the Dual-Beam FIB, FEI Helios Nanolab 600i (FEI Company, Hillsboro, OR, USA) at 5 kV and 0.17 nA.

### 2.7. Acarbose Quantification Using High-Performance Liquid Chromatography (HPLC)

For the quantification of acarbose, 2 × 1 mL of supernatant was collected and stored at −20 °C until methanol extraction and high-performance liquid chromatography (HPLC) was performed. Methanol extraction was performed after centrifugation of 1 mL supernatant for 2 min at 14,000 rpm to discard remaining solid particles. The supernatant was mixed at a ratio of 1:5 with methanol (100% *v*/*v*) and vortexed before centrifugation at 14,000× *g* for 2 min to remove the precipitate, and the methanol phase was transferred into HPLC vials. For the analysis, an Agilent 1100 HPLC system (G1312A Binary Pump, G1329A ALS autosampler, G1315A diode-array detector (DAD)) using a Hypersil APS-2 column, Thermo Fisher Scientific (Waltham, MA, USA) (125 × 4 mm, particle size: 3 μm) at 40 °C. As mobile phase containing 26% phosphate buffer (0.62 g·L^−1^ KH_2_PO_4_ and 0.38 g·L^−1^ K_2_HPO_4_ × H_2_O) (solvent A) and 74% acetonitrile (solvent B) was used with an isocratic flow rate of 1 mL min^−1^. A measure of 40 μL of each sample was injected and separated over 15 min. Acarbose was detected at 210 nm using a DAD detector, and a standard curve allowed the acarbose quantification over the peak area.

### 2.8. RNA Isolation

During cultivation, samples of 2 × 1 mL were taken and subsequently frozen in liquid nitrogen and stored at −80 °C until further isolation using the Zymo, RNA Clean & Concentrator R1014 and Quick-DNA Fungal/Bacterial Miniprep Kit D6005 (Zymo Research Corporation, Irvine, CA, USA) according to the user protocol. Cell disruption was performed with 1 mL of cultivation sample at 2 × 2600 for 30 s with 5 min on ice in between. Disrupted samples were centrifuged for 4 min at 4000 rpm at 4 °C. DNaseI digestion was performed with 75 µL of digestion buffer and 5 µL of enzyme for 15 min at room temperature, and RNA was eluted with 87 µL RNase-free H_2_O. Control PCR to exclude remaining DNA was performed with genomic context binding oligonucleotides and Phusion High-Fidelity PCR Master Mix with GC Buffer (Thermo Fisher Scientific, Waltham, MA, USA) at 62 °C for 30 s and following agarose gel electrophoresis. If required, an additional DNaseI treatment of 15 min at RT was performed with 85 µL samples, 10 µL Digestion Buffer and 5 µL DNaseI. RNA elution was performed using 30 µL RNase-free H_2_O. RNA concentration and purity were analyzed using a NanoDrop 1000 spectrometer (Peqlab, Erlangen, Germany).

### 2.9. Reverse Transcription Quantitative PCR (RT-qPCR)

A Luna ^®^ Universal Probe One-Step RT-qPCR Kit (New England BioLabs, Ipswich, MA, USA) was used for RT-qPCR with 100 ng RNA according to the manufacturer’s protocol. Ninety-six-well LightCycler plates (Sarstedt, Nümbrecht, Germany) were used for the analysis with the LightCycler 96 System (Roche, Mannheim, Germany).

### 2.10. RNA-Sequencing and Data Processing

RNA quality was analyzed on an Agilent Bioanalyzer 2100 (Agilent Technologies, Santa Clara, CA, USA). For every sampling point, RNA of three biological replicates was isolated and analyzed. cDNA libraries were prepared according to [11] using the Illumina TruSeq stranded mRNA kit (Illumina, San Diego, CA, USA) and sequenced on Illumina NextSeq 500 in paired-end mode with a 2 × 74 nt read length. Reads were mapped on the reference genome (GenBank: LT827010.1). Differential gene expression analysis of triplicates including normalization was performed using Bioconductor package DESeq2 [72] included in the ReadXplorer v2.2 software [59]. The signal intensity value (*A*-value) was calculated by the log2 mean of normalized read counts and the signal intensity ratio (*M*-value) by the log2 fold change. The evaluation of the differential RNA-seq data was performed using an adjusted *p*-value (P_adj_) cutoff of P_adj_ ≤ 0.05 and a signal intensity ratio (*M*-value) cutoff of ≥1 or ≤−1 if not mentioned otherwise. Genes with an *M*-value outside this range and P_adj_ ≤ 0.05 were considered as differentially up- or downregulated.

Primary 5′-end-specific cDNA library was constructed and sequenced to include σ0507-specific TSSs. 5′-end cDNA library preparation was performed according to Busche et al. (2022) [73], purified and size-selected for fragments of approximately 100–1000 nt by gel electrophoresis, quantified and sequenced on a MiSeq system (Illumina, San Diego, CA, USA) in PE mode with 2 × 75 nt.

RNA-seq data were deposited in in the ArrayExpress database (www.ebi.ac.uk/arrayexpress, accessed on 15 February 2024) under accession number E-MTAB-13644.

### 2.11. Electrophoretic Mobility Shift Assays (EMSA)

The vector pJOE was used for protein expression with a C-terminal His_6_ tag according to Nölting et al. (2023) [4]. Protein expression was verified via SDS PAGE and LC-MS/MS using a nano-LC Ultimate 3000 (Thermo Fisher Scientific, Waltham, MA, USA), coupled with ESI-Orbitrap MS/MS QExactive Plus (Thermo Fisher Scientific, Waltham, MA, USA), in negative ionization mode.

Complemental oligonucleotides of 61 bp were ordered (Sigma Aldrich—Merck KGaA, Darmstadt, Germany), with one oligonucleotide of each pair 5′-end biotin labeled. Annealing to a double-stranded DNA fragment was performed in 30 mM of HEPES at pH 7.8 under a temperature gradient from 98 °C to 4 °C with 1 °C min^−1^. As a control, a 61 bp fragment of the luciferase coding sequence was used. Protein–DNA interaction was analyzed using a LightShift^®^ Chemiluminescent EMSA Kit (Thermo Scientific, Waltham, MA, USA), following the manufacturer’s protocol. For the reaction, 0.1 ug (3.40 µM) recombinant protein and 0.05 nM dsDNA were used. Blotting was performed on an Amersham Hybond^TM^-N^+^ nylon transfer membrane (GE Healthcare, Chicago, IL, USA) with blot filter paper (Bio-Rad Laboratories, Inc., Hercules, CA, USA) and Trans-Blot^®^ SD Semi-Dry (Bio-Rad Laboratories, Inc., Hercules, CA, USA).

## 3. Results

### 3.1. Sigma Factor Gene ACSP50_0507 Encodes a σH Homolog of Streptomyces coelicolor

Sigma factors influence a wide variety of cellular processes and form highly complex regulatory networks [74,75]. *Actinoplanes* sp. SE50/110 carries at least 78 sigma factor genes and 52 genes coding for anti-sigma factors and anti-sigma factor antagonists (GenBank: NZ_LT827010.1). Thereby, its sigma factor regulation seems to be of higher complexity than in *C. glutamicum* with 7 σ factors, or even *S. coelicolor*, with 65 sigma factor coding genes [38].

*Actinoplanes* sp. SE50/110 alternative sigma factor gene *ACSP50_0507* is strongly transcribed during the transition and early stationary growth phase, with a maximal transcript amount at the early stationary growth phase and therefore possibly influencing the cells during growth phase transition [12]. This σ B/F/G family protein (IPRO014322) of 270 amino acids features RNA-polymerase binding and −10 promoter recognition region 2 (IPRO07027), RNA-polymerase and extended-10 binding region 3 (IPRO07624) and −35 promoter element recognition region 4 (cd06171). *ACSP50_0507* is a homolog of *A. missouriensis* (*AMIS_4630*) and *S. coelicolor* (*SCO5243*), with high protein sequence identity to *SCO5243* (56% identical amino acid residues) (Figure 1). *SCO5243* codes for σH^Sc^, which is known to be involved in osmotic stress response, sporulation and morphological differentiation [21,44] and whose regulatory network was already elucidated [21,42,43,45,46,47,76]. Sequence alignment also shows an extended N-terminal σH^Sc^ without any domains. This protein region is not essential for its functionality since *sigH^Sc^* is developmentally translated into three translational products, with the shortest one missing this extended region [77]. In *S. coelicolor*, σH^Sc^ is inactivated by anti-σ factor UshX, which itself can be inactivated by anti-anti-σ factor BldG [21]. Interestingly, homologs for both genes were identified in *Actinoplanes* sp. SE50/110, with ACSP50_7560 being homologous to UshX and ACSP50_0384 to BldG, respectively (Appendix A). Interestingly, ACSP50_0284 also shows a high similarity towards BldG but was shown in the phylogenetic analysis as more distantly related and is therefore a putative paralog.

Due to gene homology and further observations described below, ACSP50_0507 was designated σH^As^ and its gene as *sigH*^As^. In *S. coelicolor*, σH^Sc^ is encoded in an operon with its anti-σ factor gene. However, in *Actinoplanes* sp. SE50/110, that is not the case. Its homolog *ACSP50_0507* is coded in an operon under the control of several promoters. Genes which are coded in the same operon code for a putative membrane protein (*ACSP50_0506*) carry a soluble N-ethylmaleimide-sensitive-factor attachment receptor (SNARE) domain and a putative secreted SDR family NAD(P)H-dependent oxidoreductase (*ACSP50_0505*).

### 3.2. The Cell Morphology Is Highly Affected by sigH^As^ Expression

*Actinoplanes* sp. SE50/110 grows as dense mycelium, surrounded by an extended mesh of hyphae (Figure 2a). The long hyphae are occasionally branched and have a uniform shape (Figure 2b). Genomic insertion of empty vector control pSETT4*tipA* without gene of interest did not influence the strain’s morphology (Figure 2c,d). Genomic insertion of an additional gene copy (pSETT4*tipA-sigH*^As^) demonstrated a drastic influence on cell morphology, featuring a dense and irregularly formed mycelium, with no surrounding mesh of hyphae (Figure 2e). The expression strain has deformed hyphae with an irregular shape and exceeded branching (Figure 2f). Higher magnification shows that these hyphae are subdivided into parts of irregular sizes and shapes, with larger bulbous sections in between (Figure 2g). The cell surface seemingly shrank and is highly wrinkled (Figure 2h). While cell morphology of the expression strain was drastically influenced, no changes could be observed with the *sigH*^As^ deletion (Δ*sigH^As^*) and gene complementation (CΔ*sigH^As^*) strains (Appendix A), making the expression strain especially interesting for further characterization.

### 3.3. The sigH^As^ Expression Strain Achieves a Two-Fold Acarbose Yield Due to Prolonged Acarbose Production during Stationary Growth

Cultivation of *Actinoplanes* sp. SE50/110 wild type, empty vector control strain pSETT4*tipA* and expression strain pSETT*tipA*-*sigH*^As^ was performed to analyze growth and acarbose production. Growth was determined by cell dry weight determination. The expression strain has similar growth towards the end of cultivation than wild-type and empty vector control strain (Figure 3a). The expression strain achieved an increasing acarbose concentration beyond stationary growth at 118 h of cultivation, whereas the wild-type (WT) and empty vector control strain pSETT*tipA* showed a stagnating acarbose concentration from this period of growth. A maximal acarbose concentration of 0.853 g·L^−1^ at 166 h of cultivation by the *sigH*^As^ expression strain was observed, whereas WT and empty vector control had ~0.471 g·L^−1^. At this time, the expression strain achieved its optimal biomass-related acarbose yield of 63.95 mg_acab_·g_cdw_^−1^, a doubling of yield as compared to 31.75 mg_acab_·g_cdw_^−1^ achieved by the WT (Figure 3b).

The *sigH*^As^ deletion strain did show a slight growth advantage, and a similar acarbose end-concentration was achieved (Appendix A). When *sigH*^As^ was reintroduced by genomic integration using pSETT4*tipA*, the complementation strain C∆*sigH*^As^ showed similar growth behavior and acarbose end-concentration as the reference strain (Appendix A).

Gene transcription analysis using RT-qPCR revealed a 2.5-fold higher transcript level of *sigH*^As^ in the expression strain (Figure 4). This unexpectedly low differential gene transcription was further examined by a gene copy-specific amplification, using 5′UTR-specific oligonucleotides. It revealed downregulation of the chromosomal *sigH*^As^ gene copy in the expression strain, thereby indicating a much higher transcript level of the gene copy on the integration vector. It further indicates negative regulation, when σH^As^ levels are increased due to the expression of a second gene copy. The relative transcript level of the deletion strain is at the lower limit of detection, and the transcriptional levels of *sigH*^As^ were apparently restored to the wild-type level by gene complementation.

### 3.4. Increased Transcription of Acarbose Biosynthesis Genes during Growth and Late Stationary Growth Phase

Comparative whole transcriptome analysis of empty vector control strain and *sigH*^As^ expression strain revealed a high influence on gene transcription in general, with varying differential gene transcription during cultivation. Interestingly, besides 739 significantly upregulated genes (M-value ≥ 1; P_adj_ < 0.01), we also observed 657 downregulated genes (M-value ≤ −1; P_adj_ < 0.01) during the growth phase (96 h). With 2975 genes, differential transcription is maximal during late stationary growth, where 1707 genes are upregulated (M-value ≥ 1; P_adj_ < 0.01) and 1268 genes are downregulated (M-value ≤ −1; P_adj_ < 0.01) (Appendix A). During late stationary growth, the highest number of regulatory genes, totaling 247, are differential transcribed.

The genes of the acarbose gene cluster were found to be upregulated during growth and late stationary growth phases at 96 h and 166 h of cultivation (Figure 5a), respectively, with a maximal average upregulation with an M-value of ~1.96 during the late stationary phase (166 h) and maximal upregulation of the acarbose export coding *acbWXY* operon with M-values up to 4.44 (*acbY*) (Figure 5b). During the growth phase (96 h), *acbABD* are the highest upregulated genes of the gene cluster, with M-values of 3.35 and 2.50 for the biosynthesis genes *acbA* (dTDP-glucose synthase) and *acbB* (dTDP-4-keto-6-deoxyglucose dehydratase) and 4.10 for the extracellular acarbose metabolism gene *acbD* (acarviose transferase). Interestingly, during the transition phase and early stationary growth, the *acb* genes are downregulated to an average M-value of −0.78. Thereby, *acb* gene transcription is influenced in similar levels, except the monocistronically transcribed *acbZ* (extracellular α-amylase) and *acbE* (α-amylase) genes. *acbZ* transcription is downregulated during all sampling points during stationary growth (118 h–166 h), whereas transcription of *acbE* remains upregulated even when *acb* genes are downregulated in general.

### 3.5. Altogether, 185 Genes Are Permanently Upregulated in the sigH^As^ Expression Mutant

Since gene overexpression resulted in a 2.5-fold increased transcription and sigma factors act as transcriptional activators, the focus was set on genes with increased transcription. Data filtering for genes with an increased transcription (M-value ≥1; P_adj_ < 0.05) at all analyzed sampling times left 185 genes.

Of these 185 genes, 20 genes belong to a putative NRPS biosynthesis gene cluster [11]. This cluster of 38 genes (*ACSP50_3026-3063*) is upregulated in the expression strain. Six genes of this putative biosynthesis cluster are among the 25 highest permanently upregulated genes (Table 1). It is suspected that biosynthesis cluster-specific regulators strengthened the cluster upregulation. The cluster contains an AraC family transcriptional regulator gene (*ACSP50_3061*) and an Lrp/AsnC family transcriptional regulator gene (*ACSP50_3031*). Both regulatory genes are affected within the expression strain. While the putative AraC family regulator gene is both up- and downregulated during cultivation, the Lrp/AsnC family regulator gene is only upregulated during the end of stationary growth. Since the transcription of both regulators is not continuously up- or downregulated, they are probably not involved in the permanent upregulation of the putative biosynthesis gene cluster alone.

Aligning with the highly affected cell morphology, the majority of maximal upregulated genes code for membrane-associated proteins or carry a signal peptide and are therefore putative secreted proteins. *ACSP50_3382* and *ACSP50_4922* code for putative actinobacterial holin proteins. Furthermore, the PadR transcriptional regulator coding gene *ACSP50_0424* and a haloacid dehalogenase family hydrolase coding gene *ACSP50_2587* are strongly upregulated genes. Among the 185 permanent upregulated genes are several genes related to transport, amino acid supply, regulation and trehalose biosynthesis (Appendix A). Interestingly, none of the five homologs of σH^Sc^ dependent genes *SCO3557, ssgB, dpsA, gltB* and *gltD* are found among the 185 permanent upregulated genes.

### 3.6. σH^As^ Binds to a Determined Promoter Region, Upstream of Genes Permanently Upregulated in the Expression Strain

*sigH*^As^ expression resulted in the permanent upregulation of 185 genes, belonging to 178 transcription units, which were used for binding motif prediction. TSSs were assigned according to a native 5′-end transcript dataset, obtained for the *sigH*^As^ expression and deletion strains. Upstream sequences of these genes, comprising −60 to +10 nucleotides (nt) relative to the transcription start site (TSS, +1), were used for the analysis using Improbizer and visualization using WebLogo 3.7.10 [60,61]. The emerging promoter motif contains a −35 region with 5′-G/CGTTTA/C-3′ and a −10 region with 5′-GGGTAN_3_T-3′. The −35 region and −10 region have an average distance of 14.6 ± 1.9 nt (Figure 6b). Interestingly, binding motif prediction with four known binding sites of the σH^As^ homolog σH^Sc^ [43,46] identified a highly similar binding motif of a −35 region with 5′-GTTTC/GC/G-3′ with an average distance of 14.2 nt to a −10 region of 5′-GGGT/CA-3′ (Figure 6c).

To demonstrate in vitro binding of σH^As^ to a promoter region of strong differentially transcribed *ACSP50_4135* (M-value 9.5 ± 1.64 × 10^−54^), C-terminal hexa-histidine tagged σH^As^ was recombinantly expressed in *E. coli* BL21(DE3) pLysS, verified by SDS-PAGE (Appendix A) and analysis of tryptic peptide digestion by Orbi-trap-MS/MS. Electrophoretic mobility shift assay (EMSA) was performed with biotinylated double-stranded DNA of the respective promoter region of −60 to +1 bp, which were determined based on 5′-end specific transcriptome data. σH^As^ binds to the predicted promoter region 5′-TGATTA-3′ (−35 region) and 5′-GGGGATTAG-3′ ((extended) −10 region) of *ACSP50_4135*, proven by the shift in labeled DNA and protein complex (Figure 7a). The addition of unlabeled specific dsDNA in excess proved the binding specificity of σH^As^ to the DNA sequence.

Since RT-qPCR indicated an autoregulatory effect, the promoter region of *sigH*^As^ was also examined for the predicted binding motif. A putative −35 region with 5′-CTTTCG-3′ and −10 region with 5′-GGCTACGGT-3′ was determined according to the TSS and tested for σH^As^ binding using EMSA (Figure 7b). Binding of σH^As^ to the biotinylated −60 to +1 sequence was confirmed and was reversed by the addition of unlabeled specific dsDNA in excess (Figure 7b). No band shift and, therefore, no unspecific binding of σH^As^ to an intergenic dsDNA sequence, lacking the presence of a promoter, were detected (Figure 7c).

## 4. Discussion

The investigation of the influence of sigma factor on cellular processes and acarbose production in particular is of interest since several studies already highlight the influence of sigma factor on growth and metabolite production in other bacteria [16,17,26]. This becomes even more promising since acarbose production in *Actinoplanes* sp. SE50/110 is coupled to growth [14]. In this work, we characterized the influence and regulatory mechanism of alternative sigma factor σH^As^ (ACSP50_0507).

Protein sequence alignment revealed high similarity to σH^Sc^ (SCO5243) of *S. coelicolor*, where its complex regulatory network is comprehensively characterized. It is a member of the alternative σHBF family and has a dual role in the transcriptional regulation of *S. coelicolor,* controlling osmotic stress response and septation of aerial hyphae into spores [21,45]. *SigH*^Sc^ transcription is activated upon heat shock and salt stress. σH^Sc^ is further developmentally regulated in the course of differentiation by BldD, alternative translation, and proteolysis, and transcription is autoregulated under osmotic stress through one of four known upstream promoters [21,44,77]. Among other sigma factors, σH^Sc^ is part of a complex regulatory network of σB^Sc^, the master regulator for osmotic and oxidative stress response in *S. coelicolor* and further σB homologues [22,80,81]. Thereby, the regulation of σB homologues is closely interconnected by promoter cross-recognition with both their networks and their own genes [43,46]. On the protein level, σH^Sc^ is negatively regulated by the anti-σ factor UshX, which itself can be inactivated by anti-anti σ factor BldG [45,82]. Interestingly, homologs of the direct interaction partners are present in *Actinoplanes* sp. SE50/110, with ACSP50_7560 homologous to *S. coelicolor* anti-σH^Sc^ UshX and ACSP50_0384 homologous to anti-anti-σH^Sc^ BldG. It is therefore highly likely that a similar regulatory cascade exists for σH^As^.

σB homologs are stress-associated alternative σ factors with diverse roles in morphological differentiation. Strikingly, the σH^As^ expression strain features a drastically deformed cell morphology. The hyphae show severe branching and have an irregular shape with visible compartmentalization. The cell surface is severely wrinkled and appears dehydrated. This is also reflected at the transcriptional level, where the majority of permanently highly upregulated genes are coding for membrane-associated or secreted proteins. Interestingly, *ACSP50_3382* and *ACSP50_4922* encode putative actinobacterial holin superfamily II proteins. Holin proteins promote cell lysis through their accumulation, but their involvement in differentiation, biofilm formation, stress response, non-lytic transport of toxins, gene transfer agents and proteins has also been shown [83,84,85,86]. Since cell morphology is highly affected and the cells appear dehydrated, the upregulation of two holin coding genes and further membrane-associated proteins aligns with the highly affected cell morphology.

σH^As^ expression leads to the global and permanent upregulation of a putative NRPS biosynthesis gene cluster, comprising 38 genes (*ACSP50_3026-3063*), of which 6 genes are among the 25 highest permanently upregulated genes. Two transcriptional regulator coding genes (*ACSP50_3061* and *ACSP50_3031*) are located within the putative biosynthesis gene cluster. Since both are not permanently up- or downregulated in the expression strain, it is not likely that any of them cause permanent cluster upregulation alone; therefore, transcriptional activation by σH^As^ is suspected. However, upstream sequences of transcriptional units within this gene cluster do not show similarity to the predicted binding motif. Thus, indirect cluster regulation by σH^As^ through the activation of further putative regulatory elements can be assumed. Within the *sigH^As^* expression strain, a maximal number of genes is transcriptionally upregulated at late stationary growth, with increasing upregulation throughout the stationary phase. During late stationary growth, the influence on regulatory genes is highest with 247 differentially transcribed regulatory genes, among them, 12 encoding σ factors, 3 anti-σ factors and 5 anti-anti-σ factors. If these factors are not themselves regulated post-transcriptionally, they will cause a broad range of secondary regulatory effects at the late stationary growth phase but probably also at the other sampled cultivation time points, where regulatory genes are also found to be up- or downregulated (Appendix A).

Since the *S. coelicolor* homolog σH^Sc^ is transcriptionally activated under osmotic stress and involved in osmotic stress response, it prompted an investigation into transcriptional indications of functional similarity of σH^As^ in response to osmotic stress [87,88,89]. One important mechanism is to prevent cell dehydration by increasing the intracellular osmotic potential. This can be facilitated by the production or import of ions, peptides, free amino acids or other osmolytes [90,91,92]. Within the *sigH^As^* expression mutant, a high number of transport-related genes are permanently significantly upregulated. Among them are peptidases, amino acid importer coding genes (*ACSP50_3876* and *ACSP50_7166*) and the operon *ACSP50_0791-0795*, coding for a putative peptide transport system similar to the known DppABCDF or OppABCDEF systems [93,94,95,96]. Such upregulation could increase the amino acid availability, which is used to decrease osmotic pressure [97,98,99]. Besides the upregulation of acarbose biosynthesis genes and an unknown NRPS biosynthesis gene cluster, the trehalose biosynthesis genes *ACSP50_5263*, coding for trehalose synthase/amylase TreS, *ACSP50_6610*, coding for maltooligosyl trehalose synthase, and the trehalose-6-phosphate synthase coding *ACSP50_3343*, are permanently upregulated. Trehalose is an organic solute that is produced or imported under osmotic stress [100,101,102]. Since no trehalose importer is currently known in *Actinoplanes* sp. SE50/110, trehalose biosynthesis is required. It was further shown that an osmolality upshift results in increased branching of hyphae in *S. venezuelae* and *S. coelicolor* [87,88], similar to the observed hyperbranching of the *sigH^As^* expression mutant.

Acarbose production in different *Actinoplanes* species is increased under a certain osmolality, which is assumed to result from increased membrane permeability and higher maltose transport [103,104]. It was further shown that by-product formation is also reduced under specific osmolality conditions [104,105,106]. Altogether, the upregulation of transport, trehalose biosynthesis and membrane-associated genes within the *sigH^As^* expression strain and its hyperbranching and dehydrated cell shape points to its putative involvement in osmotic stress response, similar to its *S. coelicolor* homolog.

A σH^As^ binding motif was predicted and has high similarity with the known binding sites of the *S. coelicolor* homolog. In this work, binding of recombinantly expressed σH^As^-His_6_ to the upstream sequence of the highest average upregulated gene *ACSP50_4135* was experimentally confirmed. Despite ectopic overexpression under the medium–strong *tipA* promoter, *sigH^As^* gene transcription is only increased 2.5-fold. Gene copy-specific RT-qPCR determined the downregulation of the chromosomal gene copy, which can be compensated by ectopic overexpression and therefore results in low overall gene upregulation. Transcriptional inactivation of the chromosomal gene copy is most likely caused by an autoregulatory effect, since σH^As^ binding to its own upstream sequence was confirmed in vitro. Interestingly, the homologous σH^Sc^ also binds within its own promoter region [20,21,45]. Despite the binding motif similarity and sequence similarity of both σ factor proteins, none of the five homologs of σH^Sc^-dependent genes are found to be permanently upregulated. Even considering their phenotypic and functional similarity, the regulatory networks of σH^Sc^ and σH^As^ seem different. It can be speculated that the expansion and functional differentiation of the σ factor family occurred during evolution after separation of the genera *Streptomyces* and *Actinoplanes*.

The expression strain achieved a doubled biomass-related acarbose yield at 166 h of cultivation. Strikingly, an increase in acarbose concentration continued during stationary growth. This is extraordinary for *Actinoplanes* sp. SE50/110, where acarbose production ends with stationary growth in the wildtype strain. σH^As^ gene deletion did not affect the acarbose yield; thus, it is not essential for acarbose production. *Acb* gene transcription is increased during the growth and late stationary phase in the *sigH^As^* expression mutant, with the exceptions of *acbZ* and *acbE*. Both gene products are not involved in the acarbose synthesis pathway itself but code for α-amylases involved in extracellular acarbose metabolism and are transcribed monocistronically [12]. Transcription of *acbZ* remains downregulated during stationary growth and *acbE* transcription remains upregulated during the start of stationary growth, when other acarbose biosynthesis genes are downregulated within the strain. In previous studies, it was shown that the upregulation of *acbE* or *acbD* did not achieve an increased acarbose concentration [13]; therefore, it is not suspected that *acbE* upregulation alone is the reason for and increased acarbose production. Since no σH^As^ binding motif was found within the *acb* gene cluster, no direct transcriptional regulation by σH^As^ is assumed. However, general *acb* gene upregulation during later cultivation times indicates at least one additional transcriptional regulatory level, highlighting the challenges to elucidate the regulation of acarbose biosynthesis in *Actinoplanes* sp. SE50/110. Acarbose production is closely connected to the cell membrane, with Acb proteins and acarbose biosynthesis located at the inner cell membrane [107]. According to the carbophore model, acarbose could function as a shuttle for the importation of malto-oligosaccharides [3,108]. Therefore, acarbose is exported and reimported as higher acarbose variants, which enables the uptake, intracellular release and metabolism of additional glucose [109,110]. A significant morphological change is likely to influence not only cell membrane-associated acarbose production but also its use as a putative carbophore.

## 5. Conclusions

We show that sigma factor engineering is a useful tool to achieve a coordinated phenotype change. Ectopic expression of the alternative sigma factor gene *ACSP50_0507* (coding for σH^As^), a homolog of σH^Sc^ of *Streptomyces coelicolor*, achieved a two-fold increased acarbose yield in *Actinoplanes* sp. SE50/110. The genetic network, regulated by σH^As^, comprises membrane-associated functions, and its cellular morphology was deformed accordingly. A binding motif for the σH^As^ promoter was determined und experimentally validated for highly upregulated genes and the σH^As^ promoter region, showing an autoregulatory function.

## Figures and Tables

**Figure 1 microorganisms-12-01241-f001:**
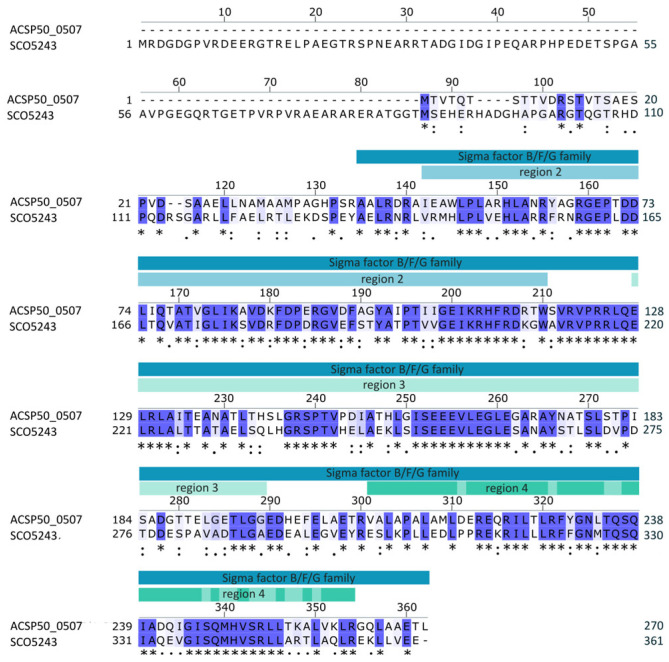
*Actinoplanes* sp. SE50/110 ACSP50_0507 protein sequence and domain annotation. Sequence alignment to σH (SCO5243) of *S. coelicolor* performed with Clustal Omega [78] and domains were predicted using InterPro [78,79]. ACSP50_0507 has a σ B/F/G family protein domain, containing three regions of different functions. Region 2 is an RNA-polymerase binding and −10 promoter recognition region, region 3 is an RNA-polymerase and extended −10-binding region and region 4 is a −35 promoter element-recognition region where DNA-binding residues are depicted in a lighter color. “*” indicates identical residues, “:“ indicates conservation between groups of strongly similar properties and “.” indicates conservation between groups of weakly similar properties.

**Figure 2 microorganisms-12-01241-f002:**
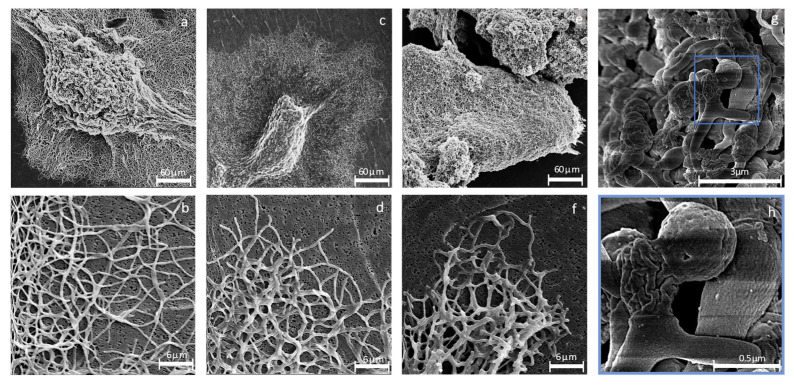
Scanning electron microscopy (SEM) of *Actinoplanes* sp. SE50/110. Strains were cultivated in maltose minimal medium for 5 days before sample preparation for microscopic imaging. The wild-type (**a**) and empty vector control strain pSETT4*tipA* (**c**) grow as large, dense mycelium, surrounded by a large web of hyphae. Hyphae of the wild-type (**b**) and pSETT4*tipA* (**d**) have a uniform shape and are occasionally branched. The morphology of the *sigH*^As^ expression strain is drastically influenced to a dense mycelium without a hyphal mesh surrounding it (**e**). Hyphae are deformed, with an irregular shape (**f**,**g**), and closer magnification of the blue marked area shows a highly wrinkled cell surface (**h**).

**Figure 3 microorganisms-12-01241-f003:**
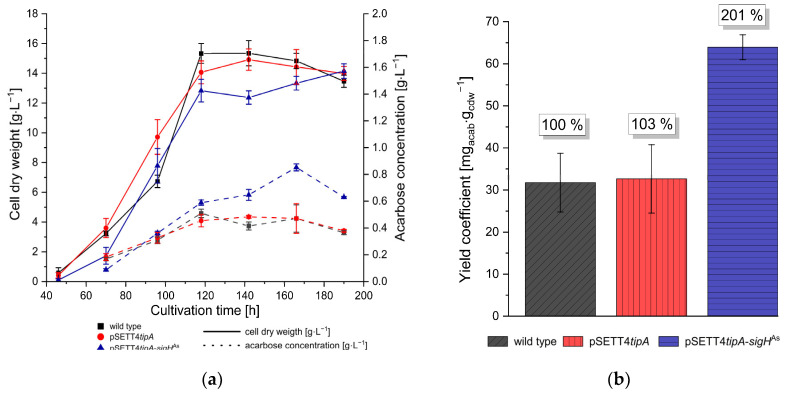
Characterization of growth and acarbose production of *Actinoplanes* sp. SE50/110 *sigH*^As^ expression and control strains. (**a**) Cell dry weight (dense lines) and acarbose concentration (bracket lines) from supernatant during cultivation in maltose minimal medium are shown for *Actinoplanes* sp. SE50/110 wild type, empty vector strain pSETT4*tipA* and σH^As^ expression strain pSETT4*tipA*-*sigH*^As^ (n = 3). (**b**) Optimal biomass-related acarbose yield of *Actinoplanes* sp. SE50/110 wild type, empty vector strain pSETT4*tipA* and expression strain pSETT4*tipA*-*sigH*^As^ at 166 h of cultivation (n = 3).

**Figure 4 microorganisms-12-01241-f004:**
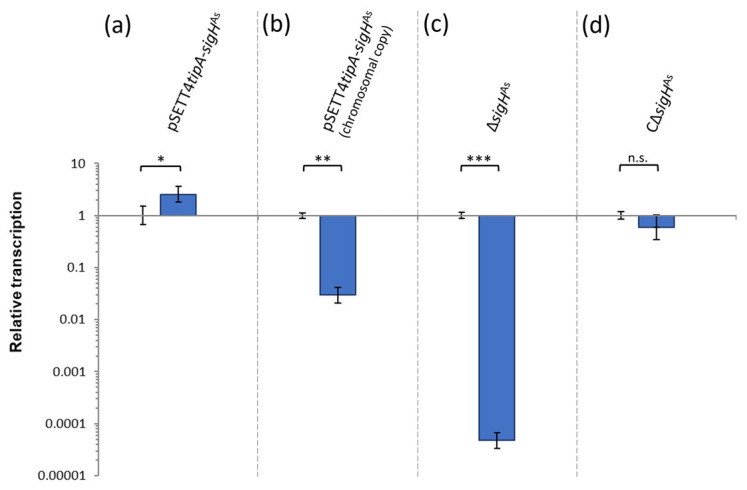
Relative *sigH*^As^ transcript amounts of *sigH*^As^ expression, deletion and complementation strains. RNA was isolated from cell pellet of 1 mL of cultivation sample at 96 h of cultivation in maltose minimal medium. RT-qPCR was performed for three biological replicates per condition with two technical replicates each. (**a**) *sigH*^As^ overexpression detected for the chromosomal and ectopic *sigH*^As^ gene copy of the expression strain (pSETT4*tipA*-*sigH*^As^) compared to the empty vector control strain. The gene expression strain (pSETT4*tipA*-*sigH*^As^) has a 2.5-fold increased transcript level. (**b**) Repression of transcription of the chromosomal gene copy. The expression strain (pSETT4*tipA*-*sigH*^As^) was compared to the empty vector control strain with 5′UTR specific oligonucleotides to differentiate between chromosomal and ectopic gene copy. The genomic integrated additional gene copy must be highly transcribed to achieve 2.5-fold upregulation detected for chromosomal and ectopic gene copy, despite the downregulation of the chromosomal gene copy. (**c**) Transcription of the deletion strain ∆*sigH*^As^ was compared to *Actinoplanes* sp. SE50/110 wild type and is at the lower limit of detection. (**d**) Gene transcription of the complementation strain (C∆*sigH*^As^) was compared to the empty vector control strain. Asterisks indicate *p*-value of a two-sided *t*-test with n.s. *p* ≥ 0.05, * *p* < 0.05; ** *p* < 0.01 and *** *p* < 0.001.

**Figure 5 microorganisms-12-01241-f005:**
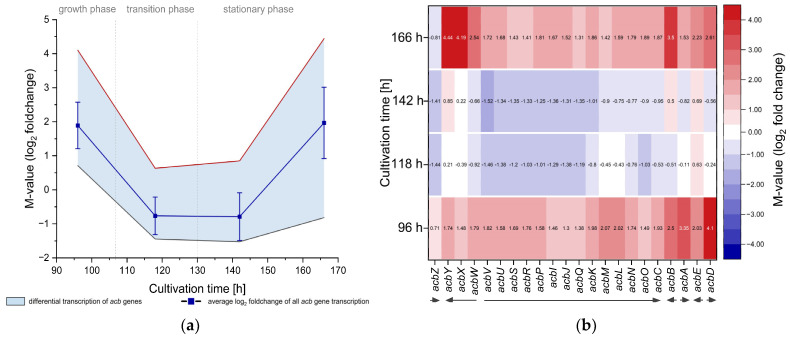
Differential transcription of *acb* genes. Transcriptome sequencing data of expression strain pSETT4*tipA*-*sigH*^As^ compared to the empty vector control strain pSETT4*tipA* during growth. (**a**) Average differential *acb* gene transcription and standard deviation of *sigH*^As^ expression strain is shown in dark blue. Minimal and maximal differential gene transcription is shown within the colored area, defined by red (maximal) or black (minimal) lines. (**b**) Differential gene transcription of single *acb* genes is shown as a heatmap with the respective M-values within the color-marked areas. Arrows indicate operon structure and gene orientation.

**Figure 6 microorganisms-12-01241-f006:**
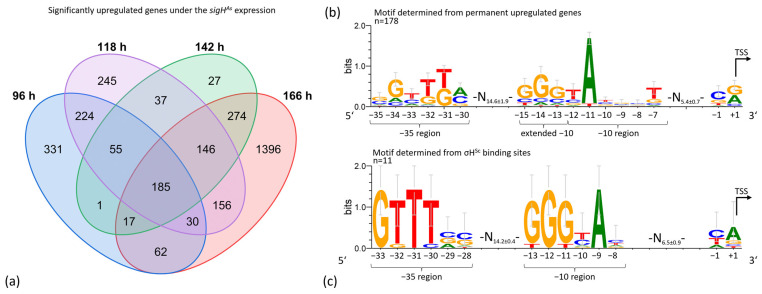
σH^As^ binding motif prediction. (**a**) Venn diagram of significantly upregulated genes (M-value ≥1; P_adj_ < 0.05) of the *sigH*^As^ expression mutant from growth to late stationary phase (96 h, 142 h, 118 h, 166 h). (**b**) Predicted binding motif of −35 and −10 region and transcription start site (TSS) for σH^As^. The motif was predicted based on upstream sequences (−60 to +10 nt) according to the TSS, determined by 5′-end specific transcript sequencing. The analysis was performed with Improbizer [61] and visualized with WebLogo 3.7.10 [60] for 178 upstream sequences of permanently upregulated genes. (**c**) Determined binding motif of the homologous σH^Sc^ of *S. coelicolor*, based on 11 known binding sites [43,46].

**Figure 7 microorganisms-12-01241-f007:**
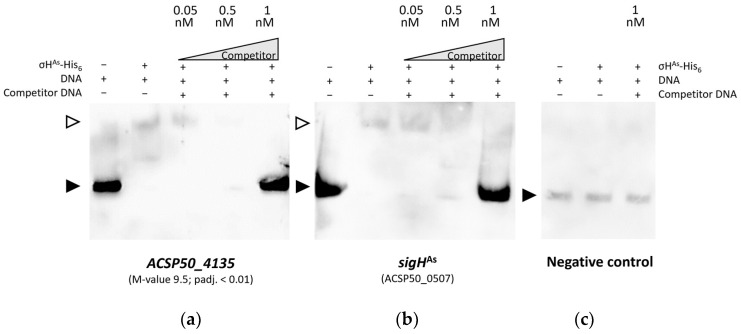
Electrophoretic mobility shift assay (EMSA) with recombinantly expressed C-terminal His_6_ tagged σH^As^. Binding was tested for upstream region of ACSP50_4135 and its own promoter. All binding reactions contained Poly (dI-dC) as a non-specific competitor and 0.05 nM of the respective biotinylated dsDNA fragment of 61 base pairs. To the reaction, 3.40 uM of protein and 0.05–1 nM of specific unlabeled competitor DNA were either added (+) or not (−). Filled arrows indicate biotinylated DNA and open arrows indicate the shifted protein–DNA complex. (**a**) EMSA with the promoter region (−60 to +1) of ACSP50_4135 confirms specific binding by σH^As^-His_6_. (**b**) Binding to the promoter region (−60 to +1) of *sigH*^As^ was also confirmed. (**c**) σH^As^-His6 does not show unspecific binding to DNA, which was confirmed by a lack of binding to the intragenic region of luciferase DNA (GenBank: EU239244.1).

**Table 1 microorganisms-12-01241-t001:** Genes of *sigH^As^* expression mutant showing the strongest average upregulation at all sampling times (96 h, 118 h, 142 h and 166 h) with M-value ≥ 1 and P_adj_ < 0.05. Genes are sorted by locus tag. The gene annotation is based on GenBank file NZ_LT827010.1 and further bioinformatic analysis.

Locus Tag	Function Annotation (NZ_LT827010.1)	Highest M-Value	P_adj_	Cultivation Time [h]
ACSP50_0208	Putative secreted protein	8.93	1.09 × 10^−14^	166
ACSP50_0424	PadR family transcriptional regulator	4.48	9.69 × 10^−15^	166
ACSP50_2587	HAD (haloacid dehalogenase) family hydrolase	5.03	2.184 × 10^−50^	142
ACSP50_2935	Protein of unknown function	4.79	3.81 × 10^−19^	142
ACSP50_3039 ^2^	oligosaccharide flippase, MATE like superfamily	4.78	2.63 × 10^−44^	142
ACSP50_3041 ^1,2^	Putative secreted protein	4.19	9.58 × 10^−23^	166
ACSP50_3042 ^1,2^	protein of unknown function (DUF 4082)	3.89	1.67 × 10^−23^	142
ACSP50_3043 ^1,2^	Glycosyltransferase family 2 protein	5.52	9.44 × 10^−21^	166
ACSP50_3047 ^2^	MbtH-like protein	5.26	6.89 × 10^−34^	166
ACSP50_3058 ^2^	Gfo/Idh/MocA family oxidoreductase	4.63	1.66 × 10^−40^	96
ACSP50_3379	Putative secreted protein	4.29	3.26 × 10^−13^	142
ACSP50_3380	Membrane protein of unknown function (DUF4235)	4.8	1.40 × 10^−12^	142
ACSP50_3381	Putative transmembrane protein (DUF 3618)	5.24	3.93 × 10^−28^	118
ACSP50_3382	Putative Actinobacterial Holin-X, holin superfamily III	5.33	1.93 × 10^−37^	118
ACSP50_3383	Uncharacterized BrkB/YihY/UPF0761 family membrane protein	4.95	7.99 × 10^−32^	118
ACSP50_3888	Putative secreted protein	8.35	1.54 × 10^−48^	118
ACSP50_4135	Putative secreted protein	9.53	1.63 × 10^−54^	118
ACSP50_4921	Putative membrane protein (DUF3618)	4.6	9.46 × 10^−16^	142
ACSP50_4922 ^1^	Putative Actinobacterial Holin-X, holin superfamily III	4.24	1.24 × 10^−12^	142
ACSP50_4923 ^1^	Protein of unknown function	4.44	1.69 × 10^−11^	118
ACSP50_5880	Putative secreted protein	10.17	3.46 × 10^−20^	118
ACSP50_6038	MFS transporter DHA2 family	6.54	2.72 × 10^−44^	166
ACSP50_6083	Putative transmembrane protein	6.98	6.50 × 10^−72^	118
ACSP50_6593	Putative transmembrane protein	4.91	1.81 × 10^−35^	142
ACSP50_7588	Protein of unknown function	5.36	4.39 × 10^−86^	96

^1^ Coded in the same operon; ^2^ members of an uncharacterized putative NRPS biosynthesis gene cluster.

## Data Availability

Data are contained within the article and Appendix A.

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
