# Peer review of "Sigma Factor Engineering in Actinoplanes sp. SE50/110: Expression of the Alternative Sigma Factor Gene ACSP50_0507 (σHAs) Enhances Acarbose Yield and Alters Cell Morphology"

_microorganisms, 2024, doi:10.3390/microorganisms12061241_

Round 1

Reviewer 1 Report

Comments and Suggestions for Authors

The authors over-expressed an alternative sigma factor in the acarbose producing Actinoplanes SE50/110. They demonstrate up-regulation of genes involved in acarbose biosynthesis. They also observed altered morphology. Genome-wide analysis of differentially expressed genes identified promoter sequences known to interact with a related sigma factor in another species. EMSAs show binding of a recombinant protein to the consensus sequence.

Overall, this is an interesting and important study. The only issue relates to the EMSA assay shown in figure 7. It would be important to demonstrate that the interacting protein is indeed the sigma factor. This could be achieved by super-shift assays with sigma HSC-specific antibodies, if available, or antibodies against a tag on the recombinant protein. Furthermore, the authors performed a competition assays with unlabeled wt DNA. They should also include a non-related DNA sequence to demonstrate DNA-sequence specificity of the interacting protein.

Comments on the Quality of English Language

The writing quality is good.

Author Response

Dear Reviewer,

We are grateful for your thoughtful feedback and have carefully considered your recommendations regarding the manuscript. Please find our dispositions to the points raised below.
Best regards

Your remarks (numbered):
1.    The only issue relates to the EMSA assay shown in figure 7. It would be important to demonstrate that the interacting protein is indeed the sigma factor. This could be achieved by super-shift assays with sigma HSC-specific antibodies, if available, or antibodies against a tag on the recombinant protein. 

Our answer:
Unfortunately, there are no specific antibodies against SigH(As) available. For using an antibody against the His6-Tag we originally planned to make an N-terminal fusion, but this construct was not viable and we therefore switched to a C-terminal His-tag fusion. However, the C-terminus (domain 4) of the sigma factor is involved in DNA-binding and therefore a band-shift can be expected to fail.
Instead, we checked our isolated protein fraction for co-precipitated transcription factors by peptide fingerprinting and mass spectrometry. We only identified three co-precipitated proteins from E. coli, the RNA-Polymerase subunits beta’ and omega as well as the ATP-dependent RNA helicase DeadD. It is not expected that these proteins bind specifically to the promoter motif. In addition, we adapted the manuscript and now mention the identified impurities in the legend of the figure S8.  

2.    Furthermore, the authors performed a competition assays with unlabeled wt DNA. They should also include a non-related DNA sequence to demonstrate DNA-sequence specificity of the interacting protein. 

Our answer:
The EMSA experiment already included several controls. We performed every binding reaction with poly (dI-dC) as an unspecific binding competitor to demonstrate DNA-sequence specificity of the interacting protein. In addition, we already used an unrelated DNA fragment (luziferase gene) in the EMSA.

Reviewer 2 Report

Comments and Suggestions for Authors

The manuscript «Sigma factor engineering in Actinoplanes sp. SE50/110: Expression of the alternative sigma factor gene ACSP50_0507 (σHAs) enhances acarbose yield and alters cell morphology» characterizes the influence and regulatory mechanism of alternative sigma factor Actinoplanes sp. SE50/110 σHAs.

Actinoplanes sp. SE50/110 is the natural producer of acarbose, an α-glucosidase inhibitor that is used in diabetes type 2 treatment. Sigma factors are transcriptional regulators that are part of complex regulatory networks for major cellular processes including growth phase-dependent regulation and stress response. Acarbose biosynthesis is dependent on growth, so sigma factor engineering is a promising tool for metabolic engineering.

Studied alternative sigma factor gene and anti-sigma factor genes in Actinoplanes sp. SE50/110 are coded in different operons. Anti-sigma factor gene is coded in an operon under the control of several promoters. Authrs show that cell morphology of the σHAs expression strain was drastically influenced, but no changes observed with the σHAs deletion and gene complementation strains. The expression of σHAs leads to a two- fold acarbose yield compared to the wild type and empty vector control strains due to prolonged acarbose production during stationary growth. Comparative whole transcriptome analysis of empty vector control strain and σHAs expression strain revealed a high influence on gene transcription in general, with varying differential gene transcription during cultivation. The genes of the acarbose gene cluster were found to be upregulated during growth and late stationary growth phases. Among the 185 permanent upregulated genes are several genes related to transport, amino acid supply, regulation and trehalose biosynthesis, but none of the five homologs of S. coelicolor sigma factor dependent genes are found among the 185 permanent upregulated genes. These 185 permanent upregulated genes belong to 178 transcription units, which authors used for binding motif prediction. Electrophoretic mobility shift assay shows binding of σHAs with predicted binding motif - promoter region of strong differentially transcribed ACSP50_4135, and with promoter region of σHAs.

The manuscript is clear, relevant for the field and presented in a well-structured manner. The cited references include recent publications (within the last 5 years) and are relevant. It does not include an excessive number of self-citations. The manuscript scientifically sound and the experimental design is appropriate to test the hypothesis. The manuscript’s results could be reproduce based on the details given in the methods section. The figures and tables are appropriate and properly show the data.

Only one notice to the figures. It would be more convenient for understanding if there will be more information on the picture (like legend for example). There are enough data in the caption for the figure, but figure 4 and figure 6 absolutely not understandable without reading.

And the one question regarding to the figure 7 – why the amount of free DNA looks lager than in complex with σHAs in the bands?

The conclusions consistent with the evidence and arguments presented.

Author Response

Dear Reviewer,

We are grateful for your thoughtful feedback and have carefully considered your recommendations regarding the manuscript. Please find our dispositions to the points raised below.

Best regards

Your remarks (numbered):
1.    Only one notice to the figures. It would be more convenient for understanding if there will be more information on the picture (like legend for example). There are enough data in the caption for the figure, but figure 4 and figure 6 absolutely not understandable without reading.

Our answer:
We have carefully implemented your recommendations regarding the manuscript and incorporated your suggestions. As you suggested, figure 4 and figure 6 were modified by adding text labels to the figures for an easier understanding. In figure 4, the legend was modified accordingly.

2.    And the one question regarding to the figure 7 – why the amount of free DNA looks lager than in complex with σHAs in the bands?

Our answer:
Thank you for the question. In EMSA, a lower signal of the shifted band is observed frequently and generally explained by a less focused band of the DNA-protein complex. In addition, the accessibility of the biotin-tag by streptavidin might lower signal intensity, too.

Round 2

Reviewer 1 Report

Comments and Suggestions for Authors

the authors have addressed previous concerns and the manuscript is acceptable.